# Beliefs about harms of cigarette smoking among Norwegian adults born from 1899 to 1969. Do variations across education, smoking status and sex mirror the decline in smoking?

**Tord Finne Vedøy***, **Karl Erik Lund**

Department of Alcohol, Norwegian Institute of Public Health, Tobacco and Drugs, Oslo, Norway

* tord.vedoy@fhi.no

## Abstract

**Data Availability Statement:** The data set is owned by a third party (Statistics Norway) and cannot be shared by us directly. However, the data

### Background and aim

Smoking is one of the most important causes of socioeconomic disparities in morbidity and mortality. The aim of this study was to examine if beliefs about harms of smoking differed across gender, smoking status and education among Norwegian adults born between 1899 and 1969.

### Methods

Using data from a nationally representative survey of smoking habits and a multinomial logit/negative binomial two-stage hurdle model design, we examined (first hurdle) the associations between birth cohort, gender, education and smoking status and four beliefs about cigarette smoking: i) smoking is not harmful, ii) do not know if smoking is harmful, iii) any number of cigarettes per day (CPD) is harmful and iv) smoking more than a given nonzero number of CPD is harmful, and (second hurdle) the predicted number of CPD that could be smoked without causing harm (from outcome iv).

### Results

The probability of believing that smoking was not harmful was close to zero, regardless of birth cohort, sex, education and smoking status. The probability of not knowing if smoking was harmful decreased from around 0.7 to almost zero across cohorts. The probability of believing that smoking more than zero CPD was harmful increased from less than 0.1 to around 0.7, while the probability of believing that there is some safe level of smoking increased with cohorts born from 1900 to 1930 before declining. Respondents with primary/secondary education consistently believed smoking to be less harmful compared to respondents with tertiary education, but cohort trajectories were similar.

is freely available to all researchers from the Norwegian Centre for Research Data and can be ordered free of charge at https://search.nsd.no/en/study/39e1c69e-aec1-44f5-963c-edaeb340c668. Creating a user profile and logging in is required.

**Funding:** The author(s) received no specific funding for this work.

**Competing interests:** The authors have declared that no competing interests exist.

## Discussion

The similar birth cohort trajectories in beliefs about the harms of smoking do not support the idea that Norwegian adults with lower education has had qualitatively different beliefs about the harmfulness of smoking compared to those with higher education. The persistent and large socioeconomic gradient is likely a result of other factors.

## Introduction

In most developed countries, cigarette smoking has declined substantially over the last decades, primarily due to lower smoking prevalence in successive birth cohorts [1–6]. This decline has been markedly different among men and women, and among groups with different socio-economic position (SEP) [2, 4–7]. Moreover, at the time when smoking prevalence began to decline, the rate of decline was faster among higher educated men compared to other groups [6–8]. In Norway, the prevalence of daily smoking decreased from around 20 to five percent among men and women with tertiary education in the period 1996 to 2016. Among men/women with primary education, daily smoking prevalence decreased from 46/42 to 25/22 percent in the same period [9]. This is in line with numbers from most other Northern European countries and the United States [10–13].

The persistent and large differences in smoking between SEP-groups is one of the most important causes of socioeconomic differences in morbidity and mortality today [14–16] and projections of smoking patterns and health outcomes for the year 2040 predict that smoking will continue to be a leading cause of mortality, socioeconomic differences in mortality [17].

The unequal rate of increase, and later decrease, in cigarette smoking between men and women, and groups with different SEP in the 20th century has been explained as a process of diffusion whereby resourceful groups adopt new practices or products (innovations) more quickly than others [18–20]. As argued by de Walque (2010), one such innovation could be the emergence of new information about the detrimental health effects from smoking from the 1960s and onwards, presented in scientific reports such as the U.S. Surgeon General's report on smoking and health from 1964 [21], but also in articles published in the general press, for example "Cancer by the Carton" published in readers Digest as early as 1952 [22]. Accordingly, health information is a central element of tobacco control policies [23] and the object of several articles (4, 11, 12 and 20) in the WHO Framework Convention on Tobacco Control [24].

Theoretical models of health behaviour, such as the theory of planned behaviour [25] and the Health Belief Model [26], argue that beliefs about risks play a key role for people's health related choices, including smoking [27]. Health information may affect behaviour in different ways, for example by altering attitudes or beliefs about smoking [28]. Moreover, models that aim at explaining how intentions and beliefs affect behaviour, such as the Health Belief Model [26] and Cognitive Dissonance theory [29] assume that the relationship between intentions, beliefs and behaviour is, to some degree, rational [30].

People with higher education are generally better informed about the harms of cigarette smoking [31–34], and several studies argue that there is a causal association between education and most health behaviours, including smoking [35–37], although some studies argue that the effect of education on smoking is not a result of being better informed, but partly or wholly a result of other "third" variables, for example delay discounting [38–41]. The high smoking prevalence among medical doctors in the 1940s and 1950s [42, 43] and the nonconstant differences in smoking prevalence between men and women with similar levels of education [6, 44] suggest that health information alone cannot explain the persistent educational gradient in smoking.

The aims of this paper are therefore to examine A) how the following four beliefs about harms of cigarette smoking varied across gender, education, smoking status among Norwegian adults born between 1899 and 1969: i) not believing smoking to be harmful, ii) not knowing if smoking is harmful, iii) believing that smoking any number of CPD is harmful and iv) believing that smoking more than a given number of CPD is harmful, and B) the predicted harmless number of CPD from outcome iv).

This enquiry serves two purposes: First, it examines whether beliefs about harms of cigarette smoking have been stable across cohorts. Second, it sheds light on the association between beliefs about harms and smoking, in a period when smoking declined substantially, and at different rates, among men and women with different levels of education. This is of interest because informing the public about the risks from smoking has been regarded as crucial for reducing the popularity of smoking [10], even though few, if any, studies have examined the long-term associations between risk perceptions, smoking prevalence and socio demographic characteristics.

## Material and methods

### Data

Data stem from Statistics Norway's nationally representative survey of smoking habits among adults (16–74 years) conducted each year from 1973 to 2020 in accordance with the Act relating to Official Statistics. Questions about beliefs about harms of cigarette smoking was included from 1973 to 1995, with the exception of 1974, 1975, 1991 and 1993. Responses from 1995 were omitted because the question differed from previous years. The mean annual number of respondents in the included surveys was 1 972. Response rates were above 85 percent in the period 1973–1990 [45] and around 70 percent in the period 1992 to 1994.

Respondents below 25 years of age were excluded to ensure that respondents were old enough to have been able to complete at least three years of tertiary education. The total number of respondents included, aged 25–74 years, was 35 487.

### Measures

*Beliefs about harms of cigarette smoking*: Respondents were asked: *How many cigarettes do you think you could smoke per day without harming your health*? Additional response options were *Do not know* and *I do not think smoking is harmful* (see Table 1 for the number of respondents and distributions on demographic and risk perception variables).

*Smoking status* was determined by asking respondents: *Do you ever smoke*? Respondents who answered *yes* were then asked if they smoked *daily* or *occasionally*. Daily and occasional smokers were combined into current smokers.

*Education*: Respondents were asked whether they had completed nine years of compulsory education (primary), additional three years of college education (secondary) or at least three years of university education (tertiary). Respondents with primary or secondary education were combined.

Respondents were asked to name their geographic *region* of residence. Four regions could be identified across all survey years: The capital (Oslo) and surround areas, eastern Norway, southern/western Norway and northern Norway.

### Hurdle model analysis

The structure and wording of the dependent variable (beliefs about harms of smoking) provided a methodological challenge. Respondents were presented with three equally valid response options, two of which a required binary response (*do not believe smoking is harmful*

**Table 1. Descriptive statistics and variable description for hurdle model 1 and 2, men and women from 25 to 74 years, 1973 to 1994.**

| | Hurdle model 1 | Hurdle model 2 |
|---|---|---|
| | (N = 31357) | (N = 10614) |
| | Mean (sd) or percent | Mean (sd) or percent |
| **Age** | 47.4 (14.2) | 46.5 (13.8) |
| **Period/real price of cigarettes in NOK** | 58.3 (2.3) | 58.4 (2.4) |
| **Birth cohort** | 1936.1 (15.6) | 1936.9 (15.1) |
| **Risk perception Hurdle model 1** | | |
| Do not believe smoking is harmful | 1.4% | - |
| Do not know if smoking is harmful | 28.8% | - |
| Believe the harmless number of CPD = 0 | 35.9% | - |
| Believe the harmless number of CPD > 0 | 33.8% | - |
| **Risk perception Hurdle model 2** | | |
| Number of CPD that can be smoked without causing harm | - | 6.0 (5.7) |
| **Sex** | | |
| Men | 47.7% | 52.0% |
| Women | 52.3% | 48.0% |
| **Smoking status** | | |
| Daily/occasionally | 53.1% | 39.8% |
| Do not smoke | 46.9% | 60.2% |
| **Education** | | |
| Primary/secondary | 84.0% | 83.9% |
| Tertiary | 16.0% | 16.1% |
| **Region** | | |
| Oslo | 21.1% | 23.3% |
| Eastern Norway excl. Oslo and surrounding areas | 29.5% | 29.2% |
| Southern and Western Norway, and Trøndelag | 38.4% | 37.6% |
| Northern Norway | 11.0% | 9.8% |

and *do not know the number of cigarettes that can be smoked per day without causing harm*) and one which required a specification of frequency (*number of cigarettes that can be smoked per day without causing harm*). Respondents were asked to respond to only one of the three questions.

One solution to this response structure was to use a hurdle model [46] in which the first hurdle consisted of calculating the likelihood of responding to the three different outcomes i) *not harmful*, ii) *do not know*, and iii) *number of cigarettes per day (CPD) is harmful*, using a multinomial logistic regression model, and in which the second hurdle consisted of estimating the predicted number of CPD reported in the third outcome using a zero inflated count model.

However, we would argue that the responses provided in the third outcome represent two qualitatively different phenomena. An answer of *zero CPD* indicated that the respondents believed that any level of cigarette use was harmful, while providing a response of one or more CPD indicated that the respondent believed there was a risk threshold where smoking more than a given number of cigarettes was harmful. We therefore believed that a better approach was to use a hurdle model in which the first hurdle (H1) consisted of estimating the likelihood of the four discrete outcomes: i) *not harmful*, ii) *do not know*, iii) *any number of CPD is harmful (harmless number = 0)* and iv) *more than a given number of CPD is harmful (harmless number > 0)* using a multinomial logistic regression model, and in which the second hurdle

(H2) consisted of estimating the predicted number of harmful CPD from outcome (4) using a zero-truncated negative binomial regression model. Negative binomial regression was preferred to poisson regression because of overdispersion. For both H1 and H2, independent variables were *education*, *smoking status*, *geographic region*, *sex*, *birth cohort*, *age* and *real price of cigarettes* as a proxy for *period* (see below).

To examine if higher educated male cohorts regarded smoking as more harmful at an earlier point in time compared to other groups, we constructed a set of additional models for both H1 and H2 with the following interactions: 2) *cohort* X *sex*, 3) *cohort* X *education*, 4) *cohort* X *smoking status*, 5) *cohort* X *sex* X *education*, 6) *cohort* X *sex* X *smoking status* and 7) *cohort* X *sex* X *education* X *smoking status* (see S1 and S2 Tables for beta coefficients and model fit). For both H1 and H2, AIC and BIC (separately and combined) favoured the model that included a three-way interaction between *smoking status*, *birth cohort* and *sex* (model H1_6 and H2_6 in S1 and S2 Tables). Goodness of fit was tested with a series of likelihood-ratio tests. The tests indicated that adding variables increased model fit for all nested models in H1 (p ≤ 0.01, S1 Table), except for the interaction between *education* and *birth cohort* in Model H1_3 (p = 0.33). In H2, adding variables increased model fit for all nested models (p ≤ 0.02, S2 Table), except for adding *education* to the three-way interaction between birth cohort, sex and smoking status in Model H2_6 (p = 0.17).

From model H1_6 (first hurdle), we calculated adjusted predicted probabilities and 95 percent confidence intervals of responding to the four possible outcomes: 1) *not harmful*, 2) do not know, 3) *harmless number of CPD = 0*, and 4) harmless number of CPD > 0) for every $10^{th}$ birth cohort (1899,..., 1969) at all values of *sex*, *education* and *smoking status* using the margins-command in Stata 15 [47]. From model H2_6 (second hurdle), we calculated the adjusted predicted number and 95 percent confidence intervals of cigarettes that respondents believed could be smoked per day without causing harm for every $10^{th}$ birth cohort at all values of *sex*, *education* and *smoking status*. Associations were tested by calculating the marginal effects (dy/dx) of birth cohort for every $10^{th}$ birth cohort at all values of *sex, education and smoking status* for both H1 and H2 (S1 and S2 Figs).

## Age, period and cohort (APC) considerations

To examine the association between birth cohort membership and beliefs about harms of cigarette smoking, it was necessary to account for the two other temporal dimensions *survey year* and *respondents' age*. However, since each temporal dimension is a perfect linear function of the two others (*period* (survey year) = *cohort* (birth year) + *age*), simultaneous estimation of all three effects is not possible without breaking the linear relationship, most often by imposing one or more restriction(s) [48]. This is referred to as the APC identification problem [49].

Several workarounds to this problem have been proposed [50–53]. Due to the relatively short series of surveys available (1973 to 1990) we employed a variant of an Age Period Cohort Characteristic (APCC) model [52] where *age* and *birth cohort* were entered as continuous variables and period was represented by a linear trend of the real price of cigarettes. The rationale behind using *real price* was that the period variable should represent short-term exogenous shocks and that long-term variations in beliefs should be a result of inter cohort ageing (age effect) and intra cohort change (cohort effect) in line with Ryders argument that social change is primarily the result of variations between cohorts and life cycle variations within cohorts [54]. A similar model was employed by Farkas to examine age, period and cohort effects upon female employment in the US [55]. Choice of APC model and alternatives are discussed below.

## Results

In total 31 357 respondents had complete information on all relevant variables (Table 1). Of these, around 1 percent (n = 452) believed that smoking did not cause harm, 29 percent (n = 9 028) did not know if smoking was harmful, 36 percent (n = 11 263) believed that zero cigarettes could be smoked per day without causing harm, and 34 percent (n = 10 614) believed that one or more cigarette(s) could be smoked per day without harm, all survey years combined (Table 1). Furthermore, 52 percent of the respondents were women, 53 percent were current (daily or occasional) smokers and 16 percent had completed a bachelor's or a master's degree (tertiary education) or equivalent. The mean real price of a pack of 20 cigarettes was 58.3 Norwegian crowns (NOK) (range 55.4–66.3). After a drop from 66.3 NOK in 1973 (around 10 USD at that time) to 57.2 NOK in 1977, the real price was stable around 58 NOK (around 8.0 USD in 1994).

Of all respondents in the first hurdle model, 10 614 (34 percent) had provided a non-zero estimate of the number of cigarettes they believed could be smoked per day without causing harm (mean 6.0 CPD, range 1–60) and were consequently included in the second hurdle (H2). There was a higher fraction of men and smokers in the H2 sample compared to the full H1 sample.

### Estimates from first hurdle (H1)

The birth cohort specific probabilities of the four outcomes from H1 (model H1_6) are presented in Fig 1. In general, the probability (pr) of believing that smoking does not cause harm (Fig 1, first row) was below 0.10 for all birth cohorts, and close to zero for cohorts born after 1930.

The probability of answering that they did not know how many CPD caused harm (Fig 1, second row) decreased strongly across cohorts from around 0.7 to near zero inn all groups. Female non-smokers with primary/secondary education had the highest initial probability of replying that they did not know (pr = 0.79, 95% CI: 0.76–0.83), while male non-smokers (pr = 0.53, 95% CI: 0.47–0.59) and male smokers (pr = 0.53, 95% CI: 0.46–0.60) with tertiary education, and female smokers with tertiary education (pr = 0.55, 95% CI: 0.48–0.62), had the lowest initial probabilities. Both among smokers and non-smokers born up until the 1950s, respondents with primary/secondary education had a higher probability of answering that they did not know if smoking was harmful, compared to corresponding groups with tertiary education.

In contrast, the probability of believing that the harmless number of CPD was zero (Fig 1, third row) increased rapidly across cohorts. However, the probabilities were initially higher among non-smokers compared to smokers. Among the latest cohort, born in 1969, the probability of believing that any number of CPD was harmful was highest among female non-smokers with tertiary education (pr = 0.79, 95% CI: 0.76–0.82), and lowest among female smokers with primary/secondary education (pr = 0.68, 95% CI: 0.63–0.72) and male non-smokers and smokers with primary/secondary education (pr = 0.69, 95% CI: 0.65–0.73 in both cases). Group differences were largest among cohorts born in the period 1929 to 1939. For example, among women born in 1929, the probability of believing that smoking more than zero CPD was harmful varied from 0.17 (95% CI: 0.16–0.18) among male smokers with primary/secondary education and 0.20 (95% CI: 0.18–0.21) among female smokers with primary/secondary education, to 0.41 (95% CI: 0.39–0.43) among male and 0.45 (95% CI: 0.43–0.47) among female non-smokers with tertiary education.

The probability of believing that smoking one or more CPD was harmless (Fig 1, fourth row) increased with birth cohort up until the 1930s, before declining in all groups. Among

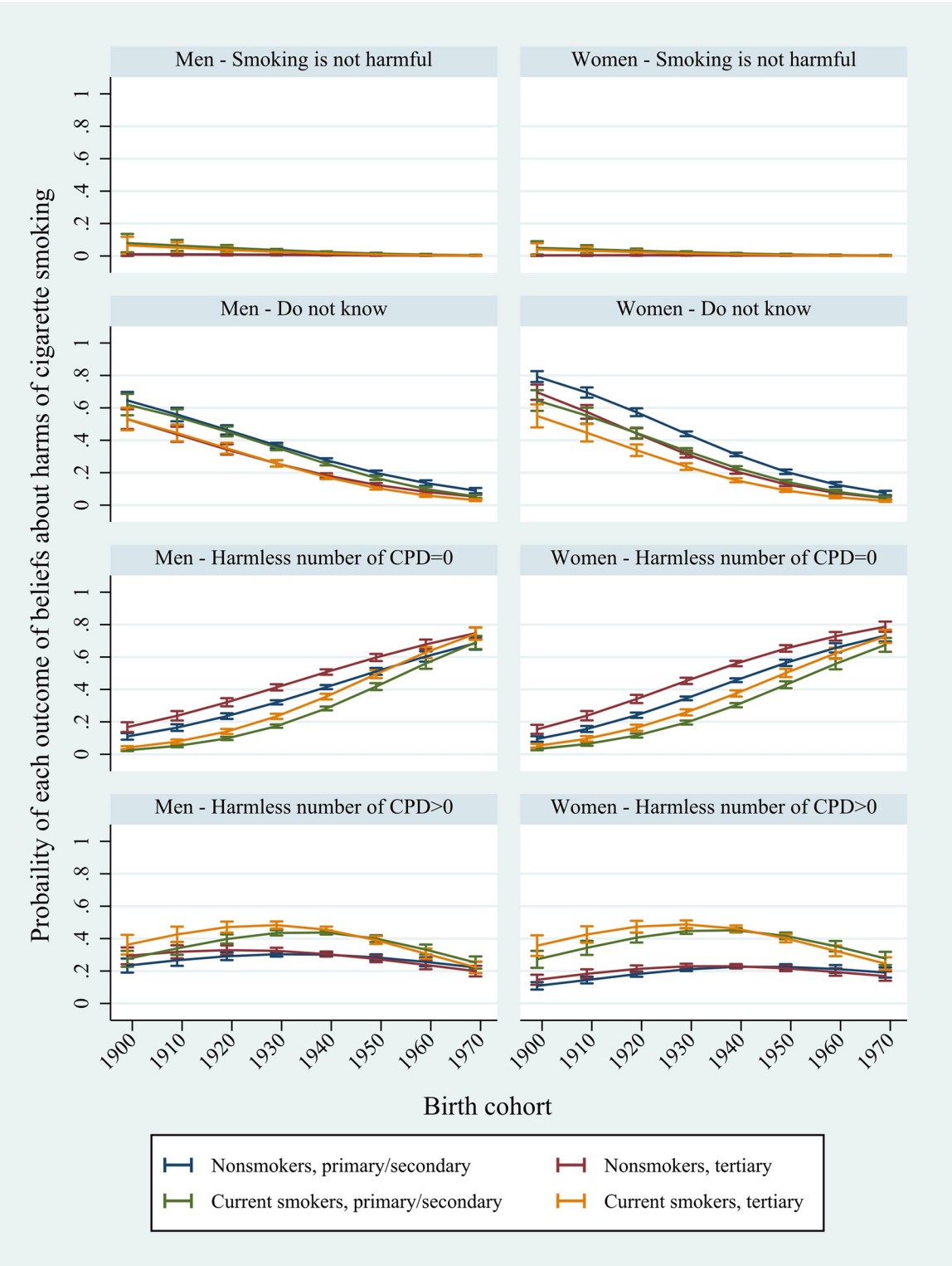

**Fig 1.** Adjusted predicted probabilities of i) believing that smoking is not harmful, ii) not knowing the safe number of CPD, iii) believing that the harmless number of CPD is zero and iv) believing that the harmless number of CPD is above zero. Male and female smokers and non-smokers with primary/secondary or tertiary education born in the period 1899 to 1969 (Model H6_1).

those born in 1929, men and women with tertiary education who smoked had the highest probabilities of believing that smoking a non-zero number of CPD was harmless (pr = 0.48, 95% CI: 0.46–0.51 for men and pr = 0.49, 95% CI: 0.46–0.51 for women). Among female non-smokers with primary/secondary education born in the same year, the probability was 0.21 (95% CI: 0.20–0.22). Among the latest cohort, the probability was highest among female smokers with primary/secondary education (pr = 0.28, 95% CI: 0.24–0.32) and lowest among female non-smokers with tertiary education (pr = 0.17, 95% CI: 0.14–0.20).

### Estimates from second hurdle (H2)

Among those who believed they could smoke one or more CPD without being harmed, the predicted number (pr_n) of cigarettes from model H6_2 decreased from 8.1 (95% CI: 6.9–9.4) among those born in 1899 to 3.7 (95% CI: 3.2–4.1) among those born in 1969, all groups combined.

As shown in Fig 2, the highest predicted number of harmless CPD was reported by male smokers with primary/secondary education born in 1899 (pr_n = 11.3, 95% CI: 9.5–13.2). In comparison, the predicted number among female non-smokers from the same cohort was around three to four CPD (pr_n = 3.1, 95% CI: 2.5–3.6 for tertiary education and pr_n = 3.9, 95% CI: 3.2–4.6 for primary/secondary education).

Among the latest cohort, born in 1969, those with highest estimate of harmless CPD was still male smokers with primary secondary education (pr_n = 4.7, 95% CI: 4.0–5.4), but the differences between groups were small. The lowest estimates were found among male non-smokers and women with tertiary education (pr_n = 2.3, 95% CI: 1.9–2.6 for men and pr_n = 2.6, 95% CI: 2.2–3.0 for women).

Among female non-smokers, the predicted harmless number of CPD remained low and stable across all cohorts, regardless of education. If we compare those born in 1899 and 1969, the predicted number decreased by 0.4 points among female non-smokers with tertiary education (from pr_n = 3.1, 95% CI: 2.5–3.6 to pr_n = 2.6, 95% CI: 2.2–3.0) and 0.6 points among female non-smokers with primary/secondary education (from pr_n = 3.9, 95% CI: 3.2–4.6 to pr_n = 3.3, 95% CI: 2.8–3.9). When comparing those born in 1899 and 1969, the largest absolute decrease in harmless CPD was found among male smokers with primary/secondary education (6.7 points, from pr_n = 11.3, 95% CI: 9.5–13.2 to pr_n = 4.7, 95% CI: 4.0–5.4), followed by male smokers with tertiary education (5.2 points, from pr_n = 8.9, 95% CI: 7.3–10.4 to pr_n = 3.7, 95% CI: 3.1–4.2) and women with primary/secondary education (5.1 points, from pr_n = 8.9, 95% CI: 7.4–10.5 to pr_n = 3.8, 95% CI: 3.2–4.4).

## Discussion

This study demonstrated that the probability of believing that smoking was not harmful was close to zero, regardless of sex, education and smoking status, among Norwegian adults born from 1899 to 1969, while the probability of not knowing if smoking was harmful decreased from around 0.7 to almost zero across birth cohorts. In contrast, the probability of believing that any amount of smoking was harmful increased from less than 0.1 to around 0.7 across birth cohorts, while the probability of believing that there was some safe level of smoking (above zero CPD) increased with cohorts born from 1900 to 1930 before declining.

The best fitting model (no interaction between *education* and *cohort*) did not support the argument that people with lower education had qualitatively different cohort trajectories for beliefs about the harms of cigarette smoking compared to those with higher education. However, compared to respondents with tertiary education, respondents with primary/secondary education were more likely to answer that they did not know how harmful smoking was

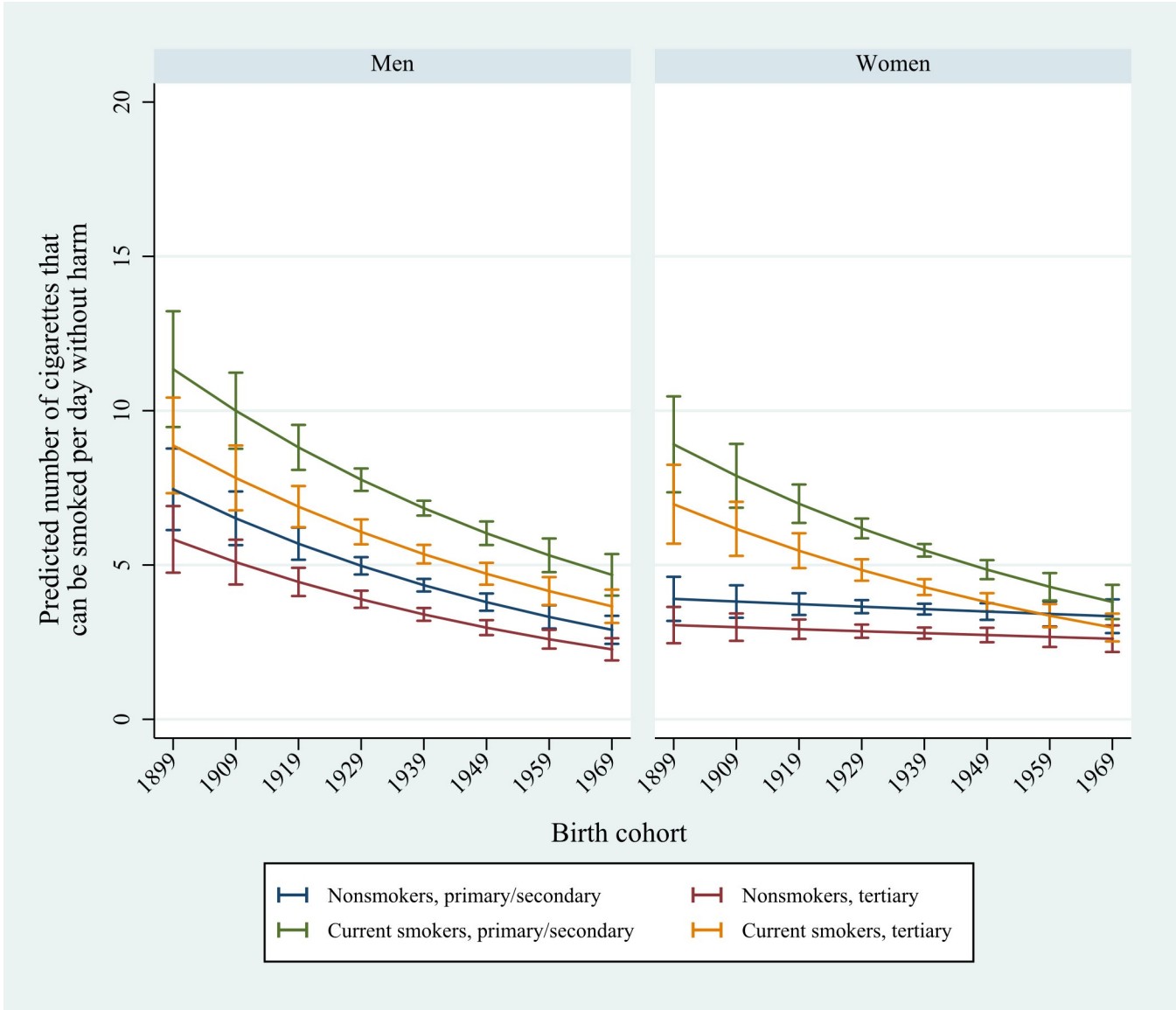

**Fig 2. Predicted number of cigarettes that can be smoked per day without causing harm.** Male and female smokers and non-smokers with primary/ secondary or tertiary education born in the period 1899 to 1969 (Model H6_2).

(hurdle 1, outcome 2), less likely to believe that more than zero CPD was harmful (hurdle 1, outcome 3) and had a higher threshold value for how many CPD they believed could be smoked without harm (hurdle 2). However, among the latest cohorts, there were little or no educational differences.

The observed convergence in beliefs about harms of smoking among people with different levels of education has not been accompanied by a similar convergence in smoking behaviour. As shown by Vedøy [2], when comparing respondents born in the 1960s and 1970s in Norway, the smoking prevalence among men and women with primary or secondary education was more than 20 percentage points higher compared to those with tertiary education. Results from the US show similar discrepancies [4]. This is in line with the idea that educational differences in health behaviours are a result of unmeasured "third" variables [38, 56]. It should be

noted that the pattern was the same even when we included *education* in the interaction term (models H1_5 and H1_7), which indicate that the similar trajectories were not a result of model specification.

Instead, the overall impression is that while education was associated with knowing that smoking was harmful (outcome 2), differences in assessing the number of cigarettes per day believed to be harmful (outcomes 3 and 4) were primarily due to smoking status. The increase in probability of believing that there is no harmless level of smoking (hurdle 1, outcome 3) took place at an earlier point in time among non-smokers compared to smokers, smokers were more likely to believe that a given number of CPD could be smoked without causing harm (hurdle 1, outcome 4) and, as shown by the results from hurdle model 2, the number of CPD they believed could be smoked without being harmed was higher. This is likely related to unrealistic optimism or self-exempting beliefs among smokers [57]. The finding that men generally believed they could smoke more CPD without being harmed compared to women (hurdle model 2) is in accordance with other studies of smoking related risk perceptions and sex [58, 59].

Differences between education groups, men and women, and smokers and non-smokers were less pronounced among respondents in late compared to early birth cohorts. However, in contrast to the discrepancy between beliefs about harms from smoking and smoking prevalence in the case of education, the increasing and converging probabilities of believing smoking to be harmful among men and women mirror the decrease and gender convergence in smoking prevalence observed in most European countries [2, 60].

From a risk communication perspective, the high over all probability (pr) of believing that smoking any number of CPD is harmful among smokers born in 1969 (pr = 0.69, 95% CI: 0.65–0.73), the low probability of answering that they do not know how harmful smoking is (pr = 0.05, 95% CI: 0.04–0.05), and the even lower probability of believing that smoking is not harmful (pr < 0.01) suggest that smokers have adopted beliefs about the health risks from smoking that are in line with health authorities and health professionals.

Similarly, the results do not support the idea that people overestimate the absolute risks of smoking to a large degree, as discussed by Krosnick et al., [61]. According Bjartveit and Tverdal, smoking one to four CPD is significantly associated with dying from ischaemic heart disease and all-causes [62]. Results from our study showed that, among all respondents born in 1969, the probability of believing that smoking a given nonzero number of cigarettes was harmless (outcome 4) was 0.23 (95% CI: 0.20–0.26), and among these, the overall predicted number of harmless CPD was 3.7 (95% CI: 3.2–4.1). This indicate that about a quarter of the respondents born in the late 1960s underestimated the risk from cigarette smoking in this study, but not by much.

In light of the substantial increase in beliefs about harms of smoking across birth cohorts shown in this study, the relatively large fraction with lower education who continue to smoke raises questions. If smokers believe they can smoke fewer and fewer CPD without being harmed, but maintain their smoking behaviour, this should lead to increased levels of cognitive dissonance [63, 64], which again could lead to fatalism [65] and an increased fraction of "hard core" smokers.

Several studies have shown that smokers with lower education have lower intentions to quit, more often believe that they will continue to smoke, are less likely to succeed in quitting and more likely to initiate smoking [2, 4, 5, 66, 67]. Moreover, a study from England found that smokers with low social grade were equally likely to try to quit smoking compared to high social grade, but that the likelihood of success among the latter group was twice as large compared to the former group [68].

This supports the idea that smokers with lower education or low socioeconomic acknowledge the risks of smoking, but that other aspects of smoking, such as stress relief, outweigh the risks. Having lower education is strongly associated with having lower disposable income and working in jobs that are associated with high levels of job strain and lower levels of job control [69]. Several studies support the idea that social and economic hardship are crucial in explain why low skilled workers and groups with lower education smoke more, and are less likely to quit, compared to groups with higher education [70, 71]. Accordingly, a study from Ireland found that education in itself only accounted for 13 percent of the total class inequality in smoking, while economic and social difficulties accounted for almost 40 percent [72], which suggests that social and economic strain are major factors for socio-economic differences in smoking, even after education is accounted for.

## Limitations

The main limitation of this study was that the results were sensitive to how we defined *period*. We used the real price of cigarettes as a proxy for short-term changes that likely affect smoking behaviour, but not primarily beliefs about harms. There are certainly other models that could have been considered, for example hierarchical age–period–cohort (mixed effects) models [51, 73] and models using the intrinsic estimator [74], both which have previously been used in tobacco research [75, 76]. However, there has been much debate about the underlying assumptions of these models [77–81]. We therefore employed a theory driven and statistically simpler model that explicitly limited the period effects to be short-term variations in price, thereby forcing temporal variations to be a result of inter-cohort (ageing) and intra-cohort change, in line with Ryders theoretical discussion of social change [54].

However, to examine how sensitive our choice of period variable was, we calculated a set of cohort profiles from a series of models (plotted in S3 Fig) where, in the left panel, *age* and *cohort* were continuous and the *period* was: i) a continuous measure of working-time equivalents in minutes for purchasing a pack of 20 cigarettes among industrial workers, ii) a continuous measure of the consumer price index (CPI) for tobacco, iii) omitted, and, in the right panel, *age* and *cohort* were dummy variables and: iv) *period* was a continuous measure of real price, v/vi) *period* were dummy variables, and the first two/last two values of *period* was constrained to 1973/1988 respectively ("classic" constrained coefficient models), and lastly, vii) *period* was recoded to be orthogonal to a time trend (Deaton normalization with fixed effects) [82]. Estimates from the main model (H1_6) was included in the left panel for comparison.

Alternatives i), iii), iv) and vii) were to a large degree similar to the best fitting model (H1_6), while ii) and v/vi) were not. In the case of ii) the discrepancies were not surprising given that CPI increases steadily over the survey period, from 47 to 66, and was therefore in conflict with our assumption that *period* should model short-term variation with no discernible trend. In the case of v/vi), the inconsistent estimates and large variations underscore the problem of arbitrarily constraining two adjacent values in an APC model.

A second limitation is that although we had data for a long series of birth cohorts, the number of survey years and age groups included were limited (see S3 Table). We do not know if risk perceptions among smokers and non-smokers in the period after 1994 were similar to what we found in the study period. However, the survey years included covered a period with rapid changes in smoking behaviour and in which the risks from smoking received much attention.

A third limitation is that long-term variations in cigarette smoking is a product of several factors besides beliefs about harms or risk perceptions, including, but not limited to, restrictions on where to smoke, price, purchasing power and social norms. Moreover, some of these

factors interact. For example, restrictions will likely affect both beliefs about harms and social norms. Some of these changes will likely be captured by variations in real price, but the endogenous nature of beliefs and risk perceptions means that we cannot claim to have isolated any causal effect of education or smoking status on beliefs about harms of smoking. Nevertheless, if differences in beliefs were an important factor for long-term educational differences in smoking, we would expect that beliefs would vary systematically with education. The results did not support this expectation.

## Conclusions

Given the emphasis on informing the public to reduce cigarette smoking, and the large differences in smoking between groups with lower and higher education, beliefs about harms of cigarette smoking should vary systematically with education. This study found that beliefs about harms of cigarette smoking were more pronounced among those with higher education, but the differences were small and did not mirror the observed educational differences in smoking prevalence across birth cohorts. From this, we argue that educational differences in information about the harms of smoking cannot be the main reason for the large socioeconomic differences in smoking observed over time.

## Supporting information

**S1 Fig. Marginal effects of birth cohort for every 10th birth cohort at all values of *sex*, *education* and smoking status in the first hurdle (H1).**
(PDF)

**S2 Fig. Marginal effects of birth cohort for every 10th birth cohort at all values of *sex*, *education* and smoking status in the second hurdle (H2).**
(PDF)

**S3 Fig. Cohort trajectories with different period specifications.**
(PDF)

**S1 Table. Regression coefficients, standard errors and p-values for models included in the first hurdle (H1).**
(PDF)

**S2 Table. Regression coefficients, standard errors and p-values for models included in the second hurdle (H2).**
(PDF)

**S3 Table. Number of respondents by birth cohort and survey year.**
(PDF)

## Acknowledgments

Data from the Norwegian Smoking Habit Survey were collected by Statistics Norway (SSB) and distributed by the Norwegian Centre for Research Data (NSD). Neither SSB nor NSD is responsible for the analyses or interpretations in this study.

## Author Contributions

**Conceptualization:** Tord Finne Vedøy, Karl Erik Lund.

**Data curation:** Tord Finne Vedøy.

**Formal analysis:** Tord Finne Vedøy.

**Investigation:** Tord Finne Vedøy, Karl Erik Lund.

**Methodology:** Tord Finne Vedøy.

**Project administration:** Tord Finne Vedøy.

**Software:** Tord Finne Vedøy.

**Supervision:** Tord Finne Vedøy, Karl Erik Lund.

**Validation:** Tord Finne Vedøy, Karl Erik Lund.

**Visualization:** Tord Finne Vedøy.

**Writing – original draft:** Tord Finne Vedøy.

**Writing – review & editing:** Tord Finne Vedøy, Karl Erik Lund.

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
