## [Decision Letter · Decision Letter 0]

3 May 2022

PONE-D-21-31872Beliefs about harms of cigarette smoking among Norwegians born from 1899 to 1969. Do variations across education, smoking status and sex mirror the decline in smoking?PLOS ONE

Dear Dr. Vedoy,

Thank you for submitting your manuscript to PLOS ONE. After careful consideration, we feel that it has merit but does not fully meet PLOS ONE’s publication criteria as it currently stands. Therefore, we invite you to submit a revised version of the manuscript that addresses the points raised during the review process.

Please address the comments provided by the reviewer in point-wise manner. 

We look forward to receiving your revised manuscript.

Kind regards,

Rajnish Joshi

Academic Editor

PLOS ONE

Journal Requirements:

3. Your abstract cannot contain citations. Please only include citations in the body text of the manuscript, and ensure that they remain in ascending numerical order on first mention.

Reviewers' comments:

Reviewer's Responses to Questions

**Comments to the Author**

1. Is the manuscript technically sound, and do the data support the conclusions?

Reviewer #1: Yes

2. Has the statistical analysis been performed appropriately and rigorously? 

Reviewer #1: Yes

3. Have the authors made all data underlying the findings in their manuscript fully available?

Reviewer #1: Yes

4. Is the manuscript presented in an intelligible fashion and written in standard English?

Reviewer #1: Yes

5. Review Comments to the Author

Reviewer #1: 1 The manuscript is written with a measurable and focused objectives using the optimum use of available data using a theoretically sound and rationalized modelling procedure. They have given the reasons for adopting hurdle model analysis and not choosing other competitive techniques (mixed effect and hierarchical models with visualization) explicitly.

2 Table -1 where the description of the variables are shown might be represented in accordance with nature of the variables (nominal to ratio) and not by mean and sd in all cases as it does not offer much information.

3 The claim of the manuscript may be toned down little bit in terms of representing the Beliefs about harms of cigarette smoking among Norwegians born from 1899 to 1969 . As they are giving a countrywide estimates using Norway’s nationally representative survey of smoking habits from 1973 to 1995 in which several exclusions are made as per age group and non response rate. It would be helpful if they offer the exact number of cohort for each year (from 1899-1969) in all 22 years data set as supplementary table .

4 some syntax errors might be corrected like in line 70/pg4 whether do they really mean a 'longer' education or a 'higher' education? Similarly line 202/pg13 the syntax( "Of the 31 357, 10 614 (34 %)") of the sentence is little cognitively overloaded sentence which may be rewritten.

5 I am particularly impressed with the limitation section where logical rationales are offered for obligation to choose one method over another.

6 A section on effect size of selected model may be added with the p values in result section . As p value is a single number which sometimes may ignore an asymmetrical effect size. The parameters showing the explanatory powers of the models with increasing complexity of models may be added in supplementary file for the interested readers.

6. PLOS authors have the option to publish the peer review history of their article (what does this mean?). If published, this will include your full peer review and any attached files.

Reviewer #1: **Yes: **Ankur Joshi

---

## [Author Response · Author response to Decision Letter 0]

15 Jun 2022

We would like to thank the reviewer and the editor for being given the possibility of submitting a minor revision of our manuscript entitled “Beliefs about harms of cigarette smoking among Norwegian adults born from 1899 to 1969. Do variations across education, smoking status and sex mirror the decline in smoking?”

We have addressed the issues regarding data availability and style requirements, and the reviewer's comments below.

Journal Requirements:

Our reply: We have read through the PLOS ONE's style requirements and believe that our manuscript meet these requirements.

Our reply: With regards to data availability, there are legal restrictions on sharing the de-identified data set. The data set is owned by a third party (Statistics Norway) and cannot be shared by us directly. However, the data is freely available to all researchers from the Norwegian Centre for Research Data and can be ordered free of charge at https://search.nsd.no/en/study/39e1c69e-aec1-44f5-963c-edaeb340c668. Creating a user profile and logging in is required. 

3. Your abstract cannot contain citations. Please only include citations in the body text of the manuscript, and ensure that they remain in ascending numerical order on first mention.

Our reply: There are no citations in the abstract

Our reply: The believe the reference list is correct and there are no changes

Review Comments to the Author

Reviewer #1: 1 The manuscript is written with a measurable and focused objectives using the optimum use of available data using a theoretically sound and rationalized modelling procedure. They have given the reasons for adopting hurdle model analysis and not choosing other competitive techniques (mixed effect and hierarchical models with visualization) explicitly.

Our reply: Thank you for your positive review of our manuscript. All changes in the manuscript are shown in red.

In addition to the changes made following your comments, we have made three changes that should be mentioned specifically.

First, we now denote the calculated probabilities from model H1_6 as “adjusted predicted probabilities” instead of “marginal mean probabilities”. Although both descriptions are used in the Stata manual, “adjusted predicted probabilities” better describe what we have calculated.

Second, in the first manuscript, the adjusted predicted probabilities calculated from model H1_6 were calculated using the over option (margins, at(cohort=(1899(10)1969)) over(smoking_status education sex)). This option divides the sample into subgroups according to smoking_status, education and sex and then calculated the probabilities of the four outcomes. After a discussion with colleagues and a close reading of the stata manual, it seems more correct to use the at option instead (margins, at(cohort=(1899(10)1969) smoking_status=(1 2) education=(1 2) sex=(1 2)). In this case, Stata uses all the data and calculates the probabilities as if all respondents were men, women, smokers, non-smokers etc. This avoids the problem of there being unobserved group differences that we were not able to control for. In practice, this change had no substantial impact on the estimates, and differences were most often around 0.01 units.

Third, we have now specifically described the higher probability of responding “Do not know” among respondents with primary/secondary education. This was briefly mentioned in the discussion, but not stated in the results-section. We have included the following in lines 241-44: 

“Both among smokers and non-smokers born up until the 1950s, respondents with primary/secondary education had a higher probability of answering that they did not know if smoking was harmful, compared to corresponding groups with tertiary education.” 

2 Table -1 where the description of the variables are shown might be represented in accordance with nature of the variables (nominal to ratio) and not by mean and sd in all cases as it does not offer much information.

Our reply: We agree with this comment and have now made a more compact Table 1 where ratio variables are described by means and standard deviations and where nominal variables are described by percentages. Ranges for the ratio variables are now only mentioned in the text.

3 The claim of the manuscript may be toned down little bit in terms of representing the Beliefs about harms of cigarette smoking among Norwegians born from 1899 to 1969. As they are giving a countrywide estimates using Norway’s nationally representative survey of smoking habits from 1973 to 1995 in which several exclusions are made as per age group and non response rate. It would be helpful if they offer the exact number of cohort for each year (from 1899-1969) in all 22 years data set as supplementary table.

Our reply: We have moderated the claims in the title by including adults in the title: “Beliefs about harms of cigarette smoking among Norwegian adults born from 1899 to 1969…”, in the abstract (Background): “The aim of this study was to examine if beliefs about harms of smoking differed across gender, smoking status and education among Norwegian adults born between 1899 and 1969” and (Discussion): “The lack of substantial educational differences in beliefs about the harms of smoking do not support the idea that Norwegian adults…” and in the first sentence of the discussion: This study demonstrated that the probability of believing that smoking was not harmful was close to zero, regardless of sex, education and smoking status, among Norwegian adults born from 1899 to 1969…

We also included the following section when discussing the limitations of the study:

“A second limitation is that although we had data for a long series of birth cohorts, the number of survey years and age groups included were limited (see S3 Table). We do not know if risk perceptions among smokers and non-smokers in the period after 1994 were similar to what we found in the study period. However, the survey years included covered a period with rapid changes in smoking behaviour and in which the risks from smoking received much attention.”

A complete table of respondents by birth cohort and survey year have been included as S3 Table.

4 some syntax errors might be corrected like in line 70/pg4 whether do they really mean a 'longer' education or a 'higher' education? Similarly line 202/pg13 the syntax ("Of the 31 357, 10 614 (34 %)") of the sentence is little cognitively overloaded sentence which may be rewritten.

Our reply: We now use higher/lower education in all cases. 

The sentence on page 13 now reads: 

“Of all respondents in the first hurdle model, 10 614 (34 %) had provided a non-zero estimate of the number of cigarettes they believed could be smoked per day without causing harm (mean 6.0 CPD, range 1-60) and were consequently included in the second hurdle (H2).”

5 I am particularly impressed with the limitation section where logical rationales are offered for obligation to choose one method over another.

Our reply: Thank you for your comment.

6 A section on effect size of selected model may be added with the p values in result section. As p value is a single number which sometimes may ignore an asymmetrical effect size. The parameters showing the explanatory powers of the models with increasing complexity of models may be added in supplementary file for the interested readers.

Our reply: Interpreting coefficients from multinomial logistic regression is difficult, even when they are exponentiated (relative risk ratios). We calculated relative risk ratios for all models, but given the small beta coefficients in some cases, some of the RRRs, especially the constants, became very large and not substantially meaningful. We therefore calculated predicted probabilities, which are much easier to interpret and take into account the other variables in the model. For the resubmitted manuscript, we also included two supplementary figures (S1 Fig and S2 Fig) which show the effect size (dy/dx) for the same groups reported in Fig1 and Fig2 at every 10th birth cohort. At the end of the “Hurdle model analysis” section we have now included the following sentence:

“Associations were tested by calculating the marginal effects (dy/dx) of birth cohort for every 10th birth cohort at all values of sex, education and smoking status for both H1 and H2 (S1 and S2 Fig).”

We agree that measuring effect sizes in multinomial logistic regression is complicated. With regards to the explanatory powers of the models we have now included a table showing goodness of fit in S1 Table and S2 Table and included the following section describing goodness of fit of the models under Material and Methods:

“Goodness of fit was tested with a series of likelihood-ratio tests. The tests indicated that adding variables increased model fit for all nested models in H1 (p<0.01), except for including the interaction between education and birth cohort in Model H1_3 (p=0.33, S1 Table). In H2, adding variables increased model fit for all nested models (p<0.02), except for adding education to the three-way interaction between birth cohort, sex and smoking status in Model H2_6 (p=0.17, S2 Table).”

Increased gof was also observed for the less complex nested models not shown in the supplementary files (S1 Table and S2 Table), where all independent variables were entered stepwise. 

We hope these changes address your concerns.

Sincerely,

Tord Finne Vedøy and Karl Erik Lund

---

## [Editor Report · Decision Letter 1]

6 Jul 2022

Beliefs about harms of cigarette smoking among Norwegian adults born from 1899 to 1969. Do variations across education, smoking status and sex mirror the decline in smoking?

PONE-D-21-31872R1

Dear Dr. Vedoy,

We’re pleased to inform you that your manuscript has been judged scientifically suitable for publication and will be formally accepted for publication once it meets all outstanding technical requirements.

Kind regards,

Rajnish Joshi

Academic Editor

PLOS ONE

---

## [Editor Report · Acceptance letter]

25 Jul 2022

PONE-D-21-31872R1 

Beliefs about harms of cigarette smoking among Norwegian adults born from 1899 to 1969. Do variations across education, smoking status and sex mirror the decline in smoking? 

Dear Dr. Vedoy:

I'm pleased to inform you that your manuscript has been deemed suitable for publication in PLOS ONE. Congratulations! Your manuscript is now with our production department. 

Kind regards, 

on behalf of

Dr. Rajnish Joshi 

Academic Editor

PLOS ONE